# Digital-Twin-Driven Intelligent Insulated-Gate Bipolar Transistor Production Lines

**DOI:** 10.3390/s24020612

**Published:** 2024-01-18

**Authors:** Xiao Zhang, Xun Liu, Yifan Song, Xuehan Li, Wei Huang, Yang Zhou, Sheng Liu

**Affiliations:** 1The Institute of Technological Sciences, Wuhan University, Wuhan 430070, China; 2022106520001@whu.edu.cn (X.Z.); xun.liu@whu.edu.cn (X.L.); 2School of Mechanical Science and Engineering, Huazhong University of Science and Technology, Wuhan 430070, China; d202280331@hust.edu.cn; 3China-EU Institute for Clean and Renewable Energy, Huazhong University of Science and Technology, Wuhan 430070, China; xhli2022@outlook.com; 4Hefei Archimedes Electronic Technology Co., Ltd., Hefei 230094, China; wei.huang@ac-semi.com; 5School of Power and Mechanical Engineering, Wuhan University, Wuhan 430070, China

**Keywords:** digital twin, insulated-gate bipolar transistors, production line

## Abstract

With the rapid development of novel energy vehicles, power generation, photovoltaics, and other industries, power electronic devices have gained considerable attention. Insulated-gate bipolar transistors (IGBTs) have been widely used in those fields. With the emergence of intelligent manufacturing concepts such as Germany’s “Industry 4.0” and China’s “Made in China 2025”, conventional manufacturing which needs to be upgraded with higher efficiency and yield is rapidly pivoting toward digitalization and intelligence. The digital twin methodology has been extensively used in various industries for constructing virtual models of physical entities, facilitating real-time data interconnection to reduce costs and improve efficiency. This study proposes a modular intelligent IGBT production line based on the digital twin. Real-time data are transmitted from a physical line to a digital line for storage and analysis. The digital line is visualized, and an intelligent management platform containing multiple functions is developed. Additionally, a process simulation database is established to obtain the optimal process parameters. Numerous quality issues that can arise during each process of IGBT packaging are addressed using a problem-solving approach based on the digital twin methodology. Consequently, this digital-twin-based IGBT intelligent production line effectively enhances yield rates and efficiency. IGBT modules with various packaging forms such as ACF, ACE, and ACD are manufactured.

## 1. Introduction

Industry 4.0 and intelligent manufacturing are the new development direction of the global manufacturing industry [1] and have had far-reaching effects on various industries [2]. As a key technology of Industry 4.0, the digital twin has been applied in many industries [3]. In his 2002 presentation for the founding of the Product Lifecycle Management (PLM) Center introducing the digital twin, Dr. Michael mentioned the real space, virtual space, and the information flow between digital twins [4]. The twin was used to simulate flight conditions precisely and assist in the preparation for flights in advance [5]. Tao et al. proposed a five-dimensional digital twin model including services and connections, which comprehensively defines the digital twin [6]. Gartner ranked the digital twin as a top 10 strategic technology [7]. Digital twin models have been used in manufacturing, energy, agricultural, and other fields [8]. In manufacturing, digital twins are involved in all stages, including production and logistics [9].

The use of digital twins in production can help simulate and validate the processes in advance. Digital twins can increase the performance and reliability of the production line because of the transfer of the large amount of real-time data [10]. Alam et al. proposed a methodology for applying digital twin technology at a sewing assembly line which reduced production line downtime and improved production efficiency [11]. Kies et al. integrated digital twin in a battery cell production leading to resource conservation and increased production efficiency [12]. Maheshwari et al. demonstrated the integration of digital twin technology with the food supply chain, showcasing its ability to accelerate productivity within the supply chain [13]. Zhang et al. proposed an intelligent digital twin system (IDTS) based on artificial intelligence and digital twin for the paper industry. By utilizing prediction models to analyze data and monitor key manufacturing indicators, they were able to enhance energy utilization and production efficiency in the paper industry [14]. Ren et al. proposed a digital-twin-based framework for production line reconstruction and monitoring of customized product manual assembly [15]. Yuan et al. proposed a digital-twin-based coupled optimization method for an electric cable production line to address coupling issues. They introduced a piecewise coupling mechanism and a multi-objective optimization algorithm for digital twin model system. Increased the productivity, line-balancing and equipment utilization, respectively [16]. Shiu et al. developed a digital twin-driven (DTD) centering process optimization system specifically designed for high-precision glass lenses, reducing process development time while improving yield rate by 20% [17]. Pei et al. proposed a digital-twin-based quality monitoring method for the production line. Parameters that can reflect quality performance are mapped into the virtual space and optimized. The process parameters are subsequently transferred back to the physical space for manufacturing [18]. Cai et al. established digital-twin-based quality management for the aircraft final assembly line. Real-time quality data are analyzed using data mining tools to improve the quality of the assembly line [19].

Power electronics have been widely applied in various industries, including electric vehicles and new energy generation [20]. Among the extensive range of power electronic devices, the insulated-gate bipolar transistor (IGBT) stands out as one of the most commonly utilized ones [21]. Numerous studies have been conducted to investigate topics related to IGBTs. For instance, Xin et al. proposed a reliable method for measuring the junction temperature of IGBTs in DC circuit breakers [22]. Wang et al. analyzed the impact of overload current on IGBT bonding wire and module reliability due to shedding [23]. Yang et al. introduced a data-driven Convolutional Neural Network (CNN)-based approach for classifying different failure modes of IGBT modules under varying PCT conditions [24]. Through finite element simulation, Fan et al. revealed that bonding lines and solder layers are vulnerable points leading to corrosion in high humidity environments within IGBT module packaging [25]. Yang et al. presented an innovative method for extracting thermal resistance and thermal capacitance to determine temperatures at critical positions, reducing model establishment time while enhancing practicality and generalization ability [26]. Furthermore, digital twin technology is gradually being applied in the field of IGBTs. Shi et al. proposed a digital twin-based method for parameter identification of an IGBT component within a three-phase inverter, enabling the separate identification of saturation voltage and forward voltage characteristics, respectively, for reverse parallel diodes [27]. On the other hand, Li et al. put forth a thermal twin modeling approach aimed at effectively monitoring device temperatures [28]. Traditional IGBT production lines are generally composed of individual working stations and operations are conducted manually. Integrating digital twin technology into IGBT production could increase production efficiency and product yield. In this paper, a digital-twin-based IGBT intelligent production line is designed and implemented. Various packaging forms of IGBT modules such as ACF, ACP, and ACE are manufactured. This advanced manufacturing system integrates physical components along with their digital counterparts and enables seamless data interaction between them for real-time transmission and analysis across various processes.

The framework of the digital-twin-based IGBT production line is depicted in Section 2. In Section 3, essential enabling technologies such as data acquisition, data transmission, visualization of the production line, and a pivotal process database are discussed. The performance evaluation of the production line is presented in Section 4. Section 5 and Section 6 provide an extensive discussion and a conclusion for this paper. 

## 2. Framework of the Digital-Twin-Based IGBT Production Line

The IGBT is available in various packaging forms based on different topologies. IGBT chips or Fast Recovery Diode (FRD) chips of various types are bonded onto a ceramic and copper layer-based direct bonded copper (DBC). Subsequently, this DBC is attached to the bottom substrate and housed to IGBT manufacturing and testing. Digital twins that enable interaction and integration between the physical world and digital world have been widely implemented in industrial fields. Many digital twin frameworks have been developed. Tao et al. proposed a digital-twin shop-floor consisting of a physical shop-floor, virtual shop-floor, shop-floor service system, and shop-floor data [29]. Miller et al. defined digital twin as an integration of varies models with connections and data stored among them [30]. Figure 1 illustrates the proposed digital-twin-based framework of the IGBT production line consisting of four components, namely the physical production line, virtual production line, data connection, and application.

A physical production line is used for realizing the digital-twin-driven production line. Manufacturing equipment for the packaging process include a die-attaching machine, vacuum reflow oven, cleaning machine, and a wire-bonding machine. Furthermore, a visual inspection machine, testing machine, programmable logic controller (PLC) and other components for function and communication are also incorporated into the production line.

The virtual production line involves the mapping of the physical production line and is connected by real-time data. Real-time data are collected from the production line, including quality data, work order information, production quantity information. These data are stored in the My Structured Query Language (MYSQL) database. Data storage, data cleaning, data optimization, and other functions are then subsequently performed. The virtual production line can analyze historical data and real-time data to obtain optimized parameters, guiding the physical production line to solve quality problems and improve product yield. The virtual line consists of three-dimensional (3D) models with high dimension precision for equipment, layout of production line, and product models. Data storage is highly linked with the digital-twin visual part that realizes data statistics, analysis, and parameters’ optimization.

The functional applications of the digital-twin production line include production management system and visualization of the production line in the virtual space. Production orders, logistic information, quality management, and other functions can be made available in the production management system by integrating MES and ERP systems. Employees from various departments can obtain information through the intelligent production line to improve communication efficiency among various departments. Problems raised during production can be solved immediately to guarantee continuous production.

The IGBT modular production line can accommodate the manufacturing of various products, packaging process routes and test execution. The IGBT production process primarily involves chip mounting, DBC and substrate welding, wire bonding, shell assembly and testing and other crucial procedures that encompass more than 10 working stations. Stations with various cycle times are equipped with one or multiple pieces of equipment to satisfy production requirements. Figure 2 displays the overall layout of the production line. The primary process equipment comprises a patch machine, vacuum reflux furnace, wire-bonding machines, testing machinery and other associated equipment. Each station is equipped with a loading and unloading feeder and the stations are interconnected through a conveyor belt to achieve automation in the production line. Additionally, nitrogen gas tanks are provided for storing both finished and semi-finished products along with carts for efficient material handling.

Figure 3 illustrates the process flow of the IGBT production line and the corresponding parts involved in each step. Initially, the DBC is sent to the laser marking station to receive a two-dimensional code at the designated position according to drawing specifications. Each individual product is assigned a unique two-dimensional code for traceability. The DBC with the marking is transported to the screen-printing station for applying a solder paste onto the chip and pin positions. The pallet with DBC is subsequently transferred to the die-attaching station. The die-attaching station consists of five attaching machines connected in a series. A nozzle extracts the chip from the cut wafer and promptly places it onto its corresponding positions on the solder paste. The installation procedure additionally encompasses the integration of Negative Temperature Coefficient Thermistor (NTC) and capacitor elements. The DBC with the mounted chip is subsequently subjected to a vacuum reflux furnace for welding. The furnace is equipped with a pre-defined temperature profile to mitigate quality issues such as the cavity rate and chip rotation. Subsequently, the DBC with the welded chip undergoes plasma cleaning at the designated station for effective cleaning. At the X-ray inspection station, meticulous examination is conducted on the cavity rate and solder paste distribution. Subsequently, the bonding process occurs at a dedicated station comprising six pieces of bonding equipment that can automatically wire various diameters and materials specified in the drawings including 5 MIL and 15 MIL wires. The bonded module subsequently proceeds to a preliminary test conducted at the quality inspection station to identify any potential defects, such as craters or scratches. Afterward, the side frames are assembled, in which glue is evenly dispensed along the edge of DBC using a dispenser followed by a shell assembly performed by a manipulator who securely fastens bolts. A precise amount of silicone gel is subsequently injected into the shell, which undergoes curing in an appropriate furnace. Both the quantity and speed of glue filling are adjusted based on specific product requirements. Finally, products that have successfully completed all processing steps proceed to an inspection station for static testing, dynamic testing, high-temperature reverse bias (HTRB), power cycling, and other reliability tests before being packaged and dispatched to customers after appearance inspections.

## 3. Key Enabling Technologies

### 3.1. Data Acquisition of the Production Line

Data are a key element in building digital twins and the digital interconnection between physical and digital layers [31]. Examples of data acquisition techniques include image-based techniques, distributed sensor systems, and wireless communication systems [32].

The IGBT production line consists of numerous pieces of production and detection equipment along with sensors that generate abundant data throughout the manufacturing process. This phenomenon includes batch information, quality metrics, equipment parameters, and production quantity data. These data are collected by sensors and, Radio Frequency Identification (RFID), for example and are transmitted to the PLC and finally uploaded to Modbus. Figure 4 shows the data acquisition architecture. Table 1 provides the essential device and sensor information. Raw materials are placed onto pallets that move from the initial station to the final station. Each station can record information of the sub-assembly by scanning QR codes, whereas each station has its unique set of equipment parameters and quality issue information. At the screen-printing station, the measuring sensor can quantify the thickness of solder paste. Subsequently, the vision sensor can identify the position of the chip after mounting to detect quality issues such as chip rotation and solder paste connectivity. The temperature sensor helps determine an optimal temperature curve within the reflow furnace. Additionally, the power sensor helps detect power parameters during wire bonding, whereas the push-and-pull sensor measures the force of bonding wires. On the completion of the side frame assembly process, the vision sensor verifies housing positioning to ensure assembly accuracy. Acquired data are processed through data cleaning, filtering, analysis, and storage. Analysis can be performed using algorithms, such as machine learning, vector machine, deep learning and simulation software (e.g., Ansys 2022 R1, COMSOL 6.0, Ansys Icepack 2022 R1) to optimize process parameters and enhance line yields. These real-time heterogeneous data can be displayed through an information management system that can provide functions such as production order management, quality management, and logistics management.

### 3.2. Data Transmission

The establishment of an IGBT production line based on the digital twin is crucial for achieving the real-time synchronization between the physical and virtual production lines. Figure 5 illustrates the implementation of data interaction and real-time driving structure in which production line devices with diverse data interfaces transmit multi-source heterogeneous data to the gateway. These data collected from sensors and other sensing devices are subsequently transmitted to PLC and embedded in the host computer through the Process Control Unified Architecture (OPC UA) communication protocol.

The transmission of historical and real-time data from the production line includes quality information, equipment status, and other pertinent parameters that facilitate thorough analysis. These data packets consist of diverse variables such as the temperature, thickness, position, force, and state, which can undergo meticulous examination using advanced machine learning algorithms such as ANN and, support vector machine, [33] along with data processing techniques. The manufacturing processes, such as reflow soldering, wire bonding, and potting are simulated and analyzed using finite element analysis. A simulation model library is developed. By integrating the results of the simulation analysis, quality issues such as warpage during reflow, bonding push force, and solidification deformation can be mitigated during the curing of quality issues. This measure reduces the iteration cycle of production line testing when improving yield and reducing costs.

### 3.3. Realization and Visualization of the Production Line

The virtual production line digital model is created using SolidWorks 2022 software, followed by the lightweight processing of the model using 3ds-MAX 2022 software. Subsequently, the processed model is imported into Unity3D to accomplish the layout of the production line as shown in Figure 6. SolidWorks, a widely adopted 3D modeling software, plays a crucial role in the design and manufacturing of various devices and products. Unity3D excels in accomplishing the production line layout and rendering tasks with finesse. Figure 7a displays the digital model of production line equipment and Figure 7b reveals the screen display of the digital production line. A novel production line management and control platform system based on C++ is developed. The platform integrated powerful functions that can display work order completion, equipment status, quality issue statistics, and operation guidance query, among others, in real time, as displayed in Figure 8. Historical data and real-time data are stored in the MYSQL database for data analysis and processing.

### 3.4. Processes Simulation of IGBT Packaging

The digital-twin-based IGBT intelligent production line encompasses more than 10 key processes including die attaching, vacuum reflow, cleaning, wire bonding, and glue curing. Every process can encounter quality issues such as chip misalignment during die attaching stage or chip tilt and excessive holes in solder pastes. Therefore, the establishment of a process simulation library can effectively address these quality problems and enhance the performance of the production line. Furthermore, optimal parameters for each process can be determined by integrating DOE tests and least square algorithms and investigating the effect of various process parameters on quality issues.

Vacuum reflow soldering is a process of joining parts by melting the solder paste through heating. The solder paste is subsequently cooled. The IGBT production line involves primary reflow soldering to attach DBC to the substrate followed by secondary reflow soldering to connect Pin with the copper substrate. During the vacuum reflow process, warpage can occur because of a mismatch in the coefficients of thermal expansion (CTE) of various materials [34]. Over-warpage can affect reliability and quality of products. Therefore, controlling and preventing warpage is crucial. Pre-warpage is widely used in industry for this purpose. To address this problem efficiently, we develop a finite element simulation model using Ansys Workbench. The simulation model was simplified by considering solely the DBC, solder paste SAC 305 and copper substrate while excluding the chips and patterns of copper. The material properties of solder SAC 305 were characterized using the Anand model. The fundamental equations of Anand are presented as follows: 

Flow equation:(1)$${\dot{\varepsilon }}_{P}=A\exp\left(-\frac{Q}{RT}\right){\left[\sinh\left(\delta \frac{\sigma }{s}\right)\right]}^{\frac{1}{m}}$$

Evolution equation:(2)$$\dot{s}=\left\{h_{0}{\left|1-\frac{s}{s^{*}}\right|}^{a}\mathrm{sign}\left(a-\frac{s}{s^{*}}\right)\right\}{\dot{\varepsilon }}_{p}$$
where
(3)$$S^{*}=\hat{s}{\left[\frac{{\dot{\varepsilon }}_{P}}{A}\exp\left(\frac{\varphi }{RT}\right)\right]}^{n}$$

Nine material constants (i.e., $A,Q/R,\delta ,m,h_{0},\hat{s},n,a,s_{0}$) are called Anand constants and can be obtained by a serious of tests. We simulate the effect of the peak welding temperature and duration on warpage and examine the deformation of prewrapped substrates during welding processes. The result reveal that pre-warpage can effectively control the warpage problem. Figure 9 displays the verification of simulation results through experiments. The deviation between simulation and experiment is within 8%, which indicates that we can use simulation to predict the pre-warpage quantity. Figure 10 displays the warpage changing with the peak welding temperature and time.

Potting and curing is a critical process in the packaging process where the module with housing is filled with silicone gel and subsequently cured in a dedicated furnace. During this curing process, the temperature rises from room temperature to 150 °C and subsequently returns to room temperature, which may cause warpage and residual stress of housing [35]. Excessive deformation can result in inadequate sealing and other associated quality issues. To find solutions for the warpage, Ansys Workbench and Moldflow are used to perform finite element analysis to simulate the curing process and injection process and determine a solution regarding warpage (Figure 10). The governing equations for the simulation are as follows [36]:

Conservation of mass:(4)$$\frac{\partial \rho }{\partial t}+\nabla \cdot \left(\rho \vec{v}\right)=0$$

Conservation of momentum:(5)$$\rho \left(\frac{\partial \vec{v}}{\partial t}+\vec{v}\cdot \nabla \vec{v}\right)=-\nabla \cdot P+\nabla \cdot \vec{\vec{\sigma }}+\rho \vec{g}$$

Conservation of energy:(6)$$\rho C_{P}\left(\frac{\partial T}{\partial t}+\vec{v\cdot }\nabla T\right)=\nabla \cdot \left(k\cdot \nabla T\right)+\eta {\dot{\gamma }}^{2}+\dot{\alpha }\Delta H$$
where ρ is density, $\vec{v}$ is velocity, t is time, $\vec{g}$ is the gravity vector, $\vec{\vec{\sigma }}$ is the stress tensor, $C_{P}$ is the specific heat, *T* is temperature, k is heat conductivity, $\dot{\gamma }$ is the shear rate, ΔH is enthalpy change, and $\dot{\alpha }$ is the cure rate. The warpage deviation for both sides remains consistent during the heating and cooling stages, as depicted in Figure 11. On one side with two injection positions, the wild lines are concentrated in the center, with one injection position they are uniformly distributed. Hence, optimizing the number of injection positions can effectively address the issue of one-side warpage quality.

The thermal characteristics of IGBT have attracted much attention because reliability is affected when the IGBT module works in a high-temperature environment [37]. To investigate the heat dissipation performance of the module, a finite element model incorporating chip distribution is established, and Ansys Workbench is used to simulate thermal resistance for various parts. Thermal resistance can be defined as follows:(7)$$R_{th}=\frac{T_{j}-T_{a}}{P}$$
where $T_{j}$ represents the junction temperature of the IGBT module, $T_{a}$ represents the ambient temperature, and *P* is the power loss of the module [38]. The thermal resistance is measured and verified using Mentor thermal resistance measurement equipment. The thermal resistance value obtained by the test equipment is the steady-state thermal resistance value obtained under the condition of constant current and voltage, and the simulation is also based on the steady-state thermal resistance simulation. The experimental findings align with the simulation results, which validates the accuracy. Figure 12 illustrates a comparison of the simulation and experimental results.

Every process in the IGBT power module packaging process requires specialized equipment with adjustable parameters to achieve optimal production outcomes. Screen printing can regulate pressure and angle, whereas the reflux furnace can modify temperature curves for preheating peak temperature, welding peak temperature, and welding time. Ultrasonic welding equipment allows adjustments in wire force, power and time. The process parameters significantly impact the product quality and are transmitted in real-time from the physical production line to the digital line. In case of any abnormal quality issues, the digital line can analyze and optimize the parameters and transmit them back to the physical line, ensuring that effective troubleshooting measures are implemented. Figure 13 illustrates the real-time optimization process.

The system facilitates the seamless real-time collection and upload of all on-site data such as quality metrics, equipment parameters, and work orders to the virtual production line. The collected data are instantly stored in a database for display on a smart screen enabling continuous monitoring of output trends with identification of nonconforming products or detecting equipment alarms. This phenomenon allows for the prompt resolution of quality issues or equipment malfunctions.

### 3.5. Digital-Twin-Based Quality Issue Solving Method

#### 3.5.1. Framework

The management of production lines typically encompasses three aspects, namely equipment, personnel, and quality. Quality is becoming an important strategy that determines company’s success and competitiveness [39]. Any occurrence of quality issues within the production line can result in abandonment. In severe cases, quality issues can even result in line shutdown, making it impossible to satisfy daily production requirements. Moreover, if delivered products exhibit quality problems, not only will product recalls be implied, but also the brand image on the client side becomes compromised and may even affect the survival of the enterprise. To address these concerns effectively, numerous quality management tools are commonly used by most companies, including production part approval process (PPAP), advanced product quality planning (APQP), failure mode and effects analysis (FMEA), and control planning. Quality management permeates throughout the entire product life cycle, encompassing research and development design stages through small batch trial productions to formal mass productions and post-sales support activities. In terms of production line inspections specifically, they are typically categorized based on processes such as incoming parts inspection, line production inspection, finished product function testing, as well as final warehouse inspections. Additionally, automobile manufacturers regularly conduct VDA audits to assess their production processes’ adherence to standards. The detection and resolution of any quality issues at an early stage are crucial for cost savings.

The types and causes of quality issues typically exhibit numerous variations. However, certain correlations and similarities can be observed. Therefore, we propose a digital-twin-based quality problem-solving method that enables the fast resolution of production line quality problems and ensures uninterrupted production. Typically, work orders are used to execute production line operations and contain crucial information such as batch numbers and materials used. This information, including quality-related data, is transmitted from the physical production line to its virtual counterpart in real time. Therefore, an extensive amount of real time and historical quality data are stored within the database. Quality issues are initially classified according to their respective processes, such as die attaching, vacuum reflow, and wire bonding. Each process stores common quality problems based on the database records. Consequently, when a quality problem arises at any station along the production line, it can be promptly identified for subsequent analysis. A fishbone diagram quality analysis method based on the digital twin was proposed to allow for a comprehensive analysis of quality issues from multiple perspectives including operators, working machines, operating parts, operating methods, and environment. The virtual production line enables the real-time acquisition of personnel information in the specific process in which quality problems occur to rule out human error as a cause if the problem persists regardless of the operator or automated station. Subsequently, relevant material information related to the quality issue is examined to determine if it contributes to the problem. If not related to materials, further investigation focuses on abnormal process parameters or changes in the processing sequence. Finally, the system performs an assessment of equipment and environmental conditions including checking for alarm information in the equipment and abnormal ambient temperature and humidity levels. Based on this analysis, the root cause of quality issues or directions for further investigation is provided. Field operators or engineers can use these recommendations to expedite problem analysis and resolution by implementing appropriate measures such as batch part replacements, parameter optimization, equipment repairs, or enhancements in operational techniques. These problem-solving approaches are subsequently integrated into the virtual production line through a combination of database utilization and machine learning techniques. Consequently, when similar problems arise in the future, automated problem-solving measures can be promptly suggested with continuous improvement in accuracy. Commonly used measures include batch part replacements, adjustment of equipment parameters, and optimization of process operations. Furthermore, if quality issues originate from improper processes, simulation data within the virtual production line can facilitate parameter optimization. Figure 14 illustrates the framework of this quality issue analysis method.

#### 3.5.2. Case Analysis

Voids occur frequently during the reflow soldering process. High void ratios can considerably influence product quality and reliability. Therefore, controlling the void rate within 1% for individual chips and within 3% for the entire assembly is essential. When encountering a void quality issue, it is crucial to promptly collect nonconforming information such as the quantity of defective products and batch details. By referring to the pre-established quality issue database, we can precisely identify the problem in the reflow process. Through an intelligent management control platform, comprehensive data on five key elements involved in the reflow soldering process (i.e., operator, equipment, incoming parts, manufacturing process and environment) can be obtained at the first time for further analysis. In terms of the operator, as vacuum reflow is an automated station, operators are solely responsible for loading and unloading tasks, thus human factors can be excluded. Regarding equipment, no abnormal alarm records have been detected from the equipment; hence, equipment-related causes can be ruled out. For incoming parts, the solder pastes used originate from different manufacturers’ brands, so material reasons can also be eliminated. As for environment, the temperature and humidity levels during production were within acceptable ranges at that time. Based on digital-twin-based quality analysis information (as shown in Figure 15), optimization of process parameters offers a viable solution for resolving and enhancing void quality issues.

Nonconforming products are closely linked to scrapping cost management. Treatment methods of nonconforming products include repair, scrapping, and concession release. All quality problem data are transmitted to the virtual production line for calculating weekly and monthly scrap costs based on processing opinions. In traditional production lines, communication between different departments such as production department, quality department, and research and development department are typically required. However, conventional communication methods such as phone calls or meetings can be time-consuming and may result in certain departments not receiving timely and relevant information, thereby impeding problem-solving efficiency. To address this issue, a digital-twin-based intelligent production line integrates an advanced information management system and facilitates effective communication channels between different departments. As a result, departments can promptly receive quality issue information and conduct a comprehensive analysis of all relevant influencing factors, thereby enabling faster resolution of quality or equipment issues while ensuring efficient production. Figure 16 displays the communication framework.

## 4. Performance of the Production Line

The digital-twin-based IGBT intelligent production line was established in a 400 m^2^ dust-free workshop. Personnel entering and exiting the facility are required to wear protective clothing and undergo electrostatic treatment. This production line incorporates an advanced packaging process that is compatible with solder paste and solder sheet technologies, enabling lead bonding of various wire diameters and materials, terminal welding, nano silver welding. Additionally, the line facilitates both dynamic and static testing of modules as well as reliability tests such as HTRB, temperature cycling (TC), and high humidity high-temperature reverse bias (H3TRB). Currently in the mass production phase, this production line can manufacture IGBT modules in various packaging forms including ACP (100A, 150A), ACF (300A, 450A), and ACE (900A), which find applications across diverse fields such as photovoltaics and novel energy vehicles for efficient electric energy conversion, as displayed in Figure 17 [40].

The implementation of a digital-twin-based intelligent production line has resulted in an improvement in product yields, such as the yield of product ACF-2 increasing from 90% to 94%. Not only have the product yields been enhanced, but also the performance characteristics of the products have been improved. Compared with products manufactured under traditional production lines, the thermal resistance of products based on digital-twin production lines has also been enhanced. As depicted in Table 2, there has been a decrease in Rth from 0.85K/W to 0.71K/W for ACP-2 and from 0.45K/W to 0.35K/W for ACF-2.

The digital-twin-based IGBT intelligent production line offers numerous advantages over traditional production lines across various aspects. Implementing a digital-twin-based production line not only enhances product yield but also streamlines paper-based operations, facilitates information sharing and optimizes processes more effectively. A detailed comparison is presented in Table 3.

## 5. Discussion

This paper presents the development of a digital-twin-based modular IGBT intelligent production line, which enables the efficient mass production of IGBT modules in various package forms and minimizes the yield loss. Digital twin technology has been widely applied in manufacturing industries for purposes such as optimizing the production line layout and intelligent scheduling. Table 4 illustrates the application of digital-twin technology in other industries. However, its applications in process modeling and quality analysis are limited. In this study, we demonstrate how the proposed digital-twin-based IGBT intelligent production line enables real-time parameter optimization and propose a novel method for analyzing quality problems based on the digital twin.

In the process of establishing the digital-twin production line, we encountered several challenges. During the modeling phase of the production line, we faced difficulties owing to a lack of equipment dimensions and unsuccessful attempts to obtain equipment drawings from the supplier, despite multiple communications. Consequently, we invested significant time in manually measuring relevant dimensions on-site to ensure an accurate model representation. Subsequently, while constructing the information management Kanban board, we dedicated considerable effort to acquiring programming skills because they were essential for platform development. Last, to achieve precise simulation results, we had to incorporate accurate material parameters and boundary conditions into our simulation model library using different software for various process simulations. We diligently acquired knowledge about these processes and software tools through extensive experimentation aimed at calibrating our simulation models and ultimately obtaining accurate results. 

Establishing a digital-twin-based intelligent production line remains challenging. Accurate identification of causes and solutions within the human–machine material loop requires the implementation of an optimized algorithm for addressing diverse quality problems. Furthermore, enhancing interdepartmental linkage mechanisms is necessary for improving communication capabilities between departments and boosting overall factory operation efficiency in the future.

The intelligent production line built upon the foundation of the company’s products and equipment and incorporates extensive product data and process parameters that are closely linked to concerns regarding data privacy and security. To ensure utmost security, we implemented varying levels of personnel permissions based on job functions. These permissions encompass a range of activities, including viewing and modifying process parameters, accessing other relevant data, and copying information. Our stringent measures effectively safeguard the safety and stability of the production line.

## 6. Conclusions

A novel modular digital-twin-based intelligent production line was designed, integrating all packaging processes including die attaching, wire bonding, and testing. A process simulation model library was established and real-time optimization of process parameters was achieved. Additionally, the digital twin approach was used to analyze and resolve issues related to nonconforming products from various perspectives, such as the operator, machine, involved parts, methods, and environment. This phenomenon enhanced interdepartmental communication efficiency and problem-solving capabilities. The production line was situated in a dust-free workshop covering an area exceeding 300 m^2^. The line enabled on-site material and parameter switching based on various product types while establishing real-time and historical databases to enhance the production line yield and rhythm. Furthermore, the intelligent production line can implement three-shift production for over 10 product types, which could effectively introduce them to major customers such as photovoltaic and automobile companies.

The future work will focus on the following aspects: (1) We aim to develop a more precise model and algorithm based on big data to predict and optimize process parameters. Additionally, it will enable more accurate prediction and resolution of quality issues. (2) Another direction is to establish digital-twin-based connectivity throughout the entire product life cycle, encompassing research and development, testing, and after-sales stages. This will facilitate perception of changes in each process by other processes and analysis of their related impacts. It will also enable the determination of all relevant parts associated with a specific process or component for achieving manufacturing process traceability.

## Figures and Tables

**Figure 1 sensors-24-00612-f001:**
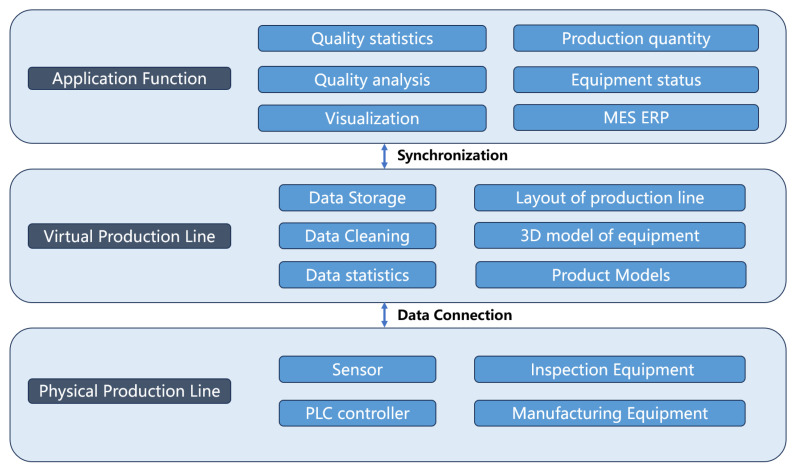
Framework of digital-twin-based IGBT production line.

**Figure 2 sensors-24-00612-f002:**
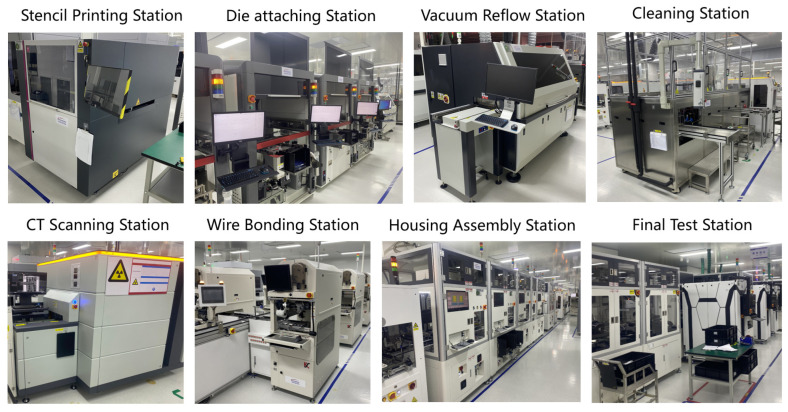
Layout of IGBT production line.

**Figure 3 sensors-24-00612-f003:**
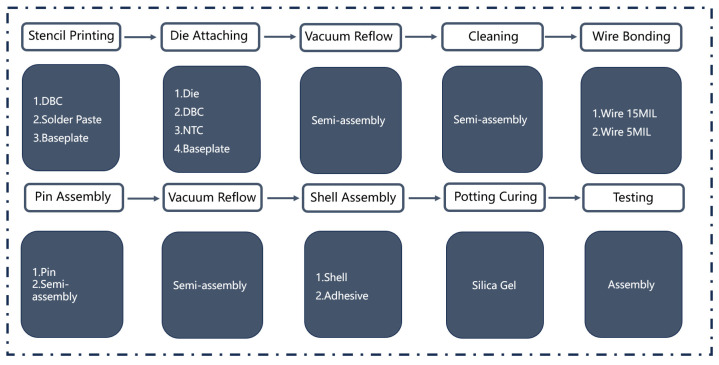
Flow chart of production line.

**Figure 4 sensors-24-00612-f004:**
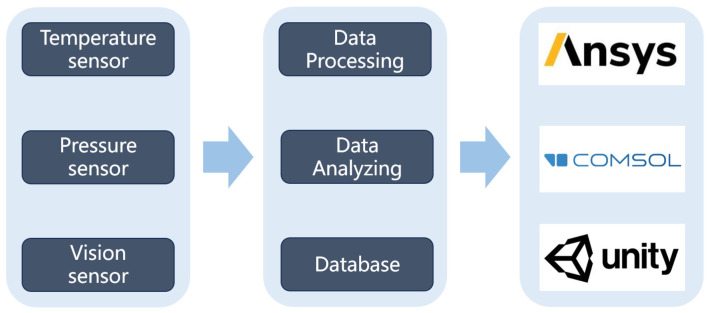
Data acquisition architecture for IGBT intelligent production line.

**Figure 5 sensors-24-00612-f005:**
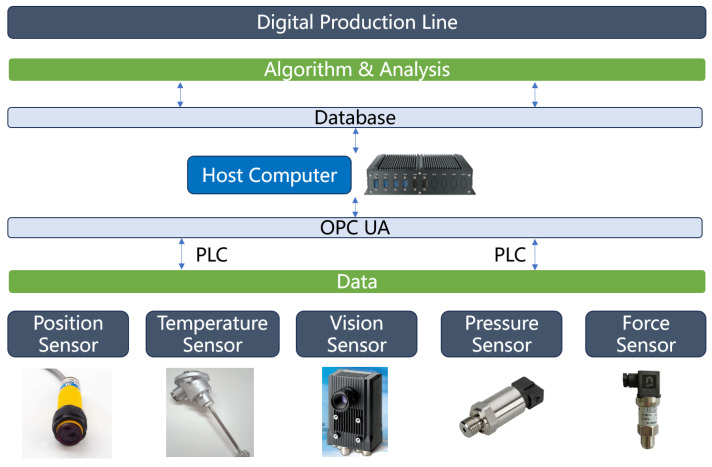
Data transmission structure.

**Figure 6 sensors-24-00612-f006:**
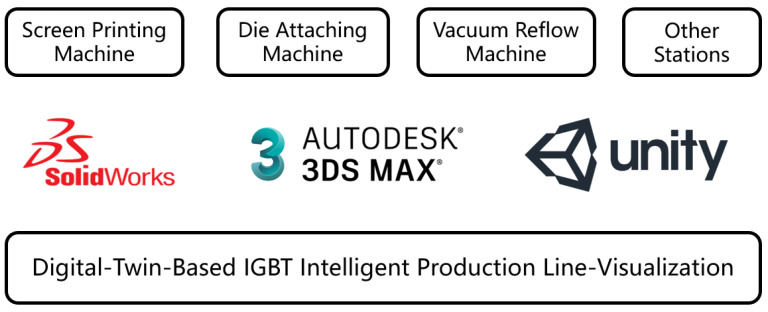
Visualization techniques of production line.

**Figure 7 sensors-24-00612-f007:**
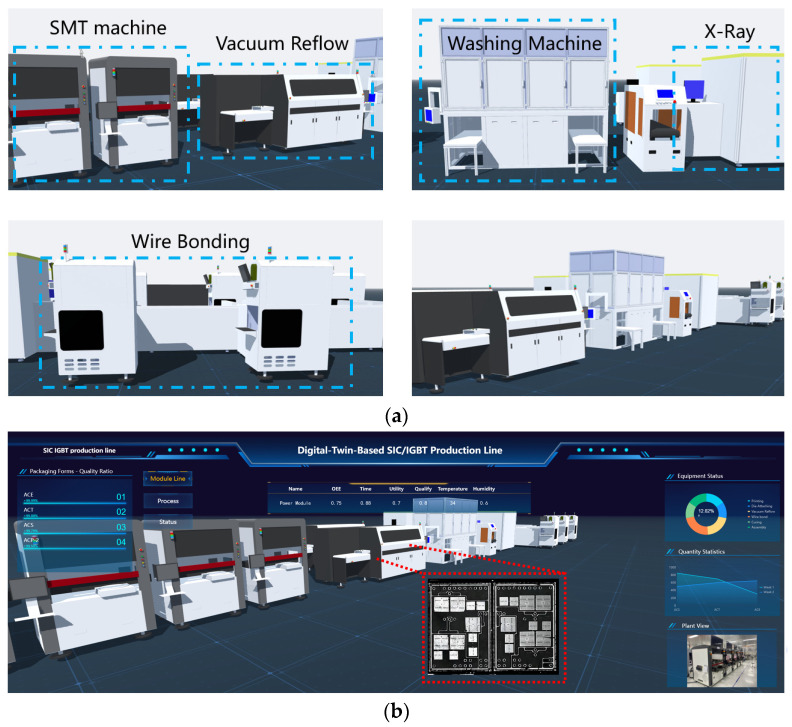
Visualization of intelligent IGBT production line (**a**) 3D model of production line. (**b**) Screen display of production line.

**Figure 8 sensors-24-00612-f008:**
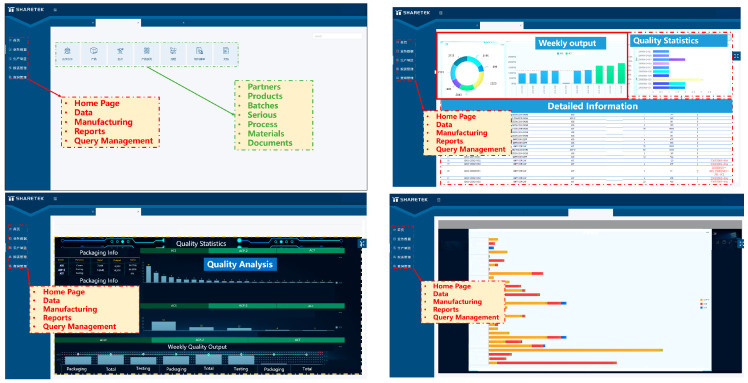
Intelligent production line management platform.

**Figure 9 sensors-24-00612-f009:**
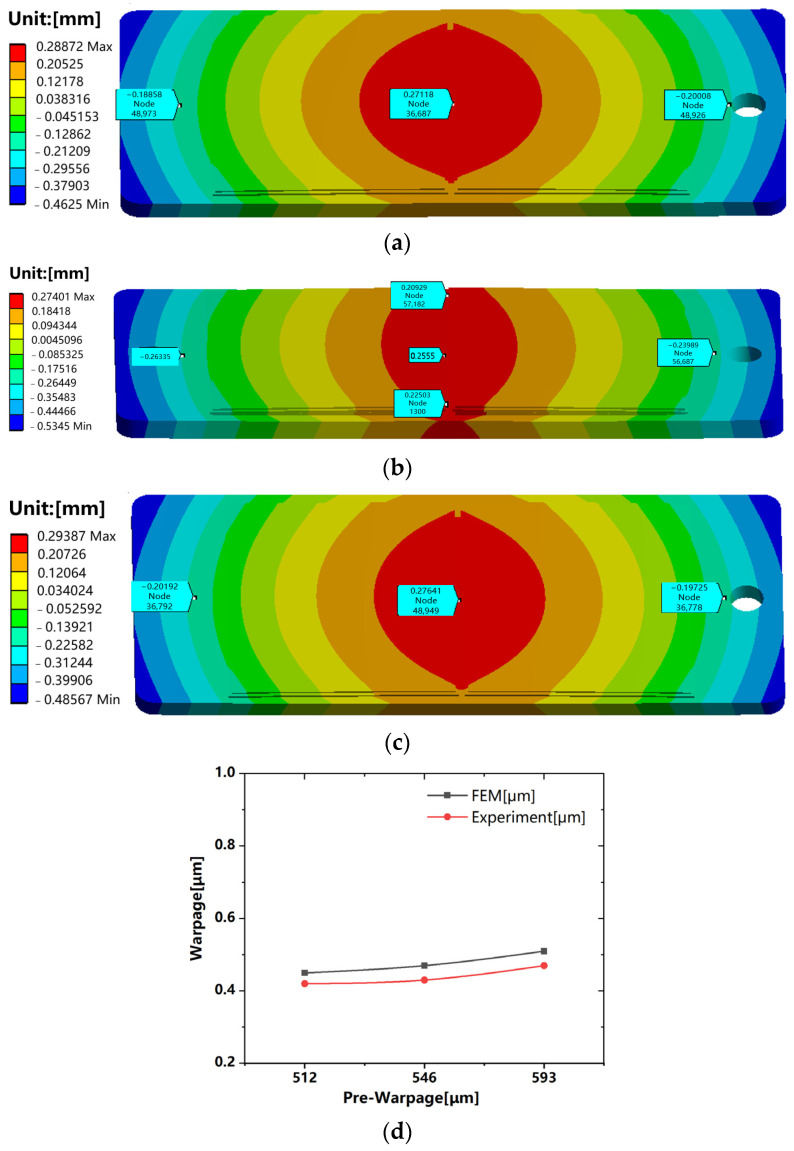
Warpage variation after vacuum reflow with various pre-warpages: (**a**) 512 μm; (**b**) 546 μm; (**c**) 593 μm; (**d**) Comparison of simulations with experimental results.

**Figure 10 sensors-24-00612-f010:**
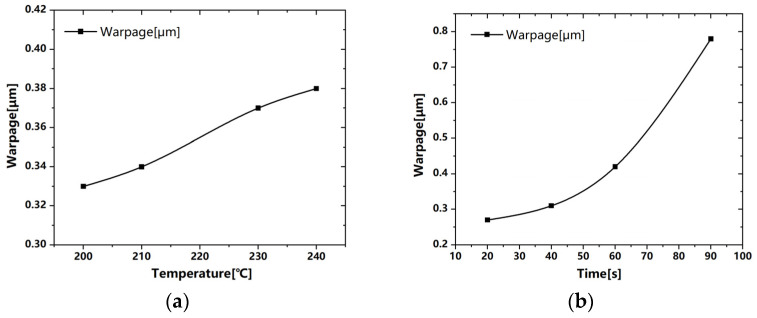
Warpage variation with changing parameters: (**a**) temperature; (**b**) time.

**Figure 11 sensors-24-00612-f011:**
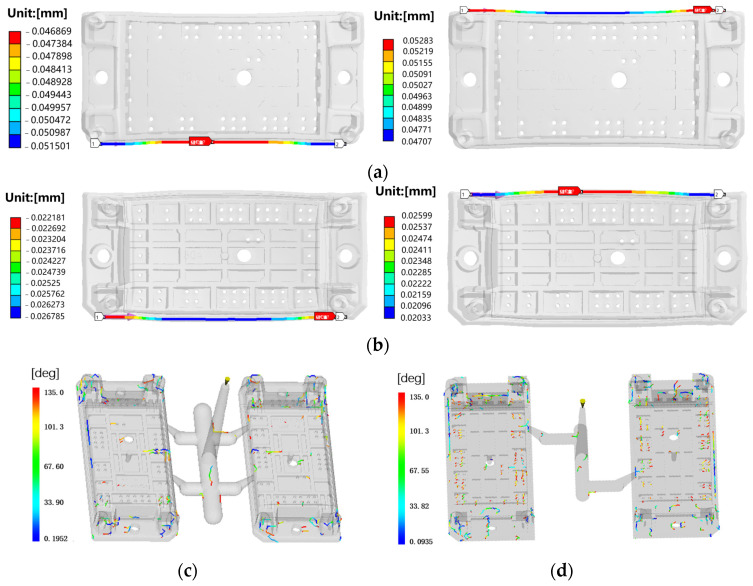
Curing process and injection molding simulation: (**a**) IGBT housing warpage. variation during heating stage; (**b**) warpage during cooling stage; (**c**) weld line positions with two injection positions; (**d**) weld line positions with one injection position.

**Figure 12 sensors-24-00612-f012:**
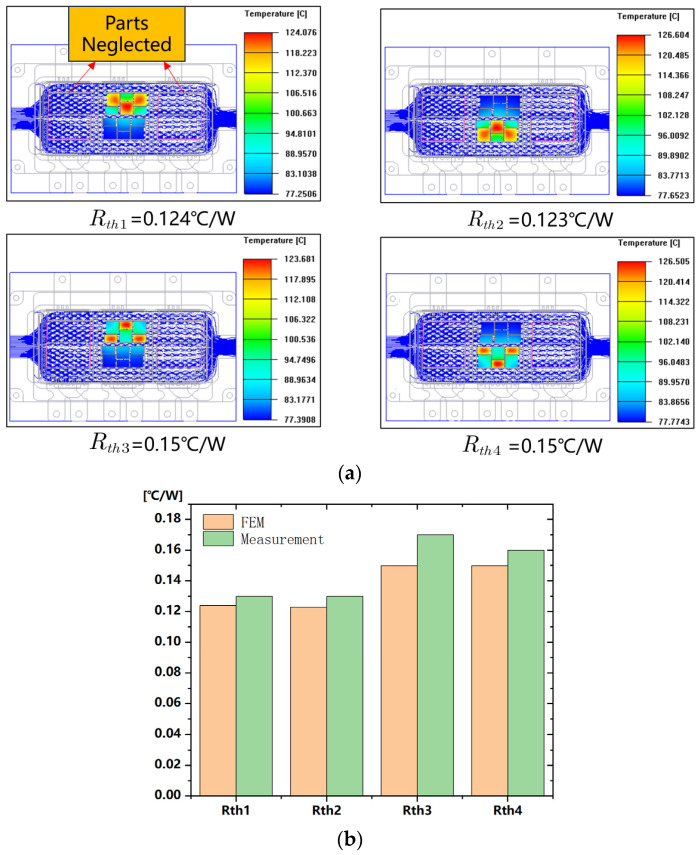
Thermal resistance measurement: (**a**) Finite element model (FEM) simulation (**b**) FEM comparison with experimental results.

**Figure 13 sensors-24-00612-f013:**
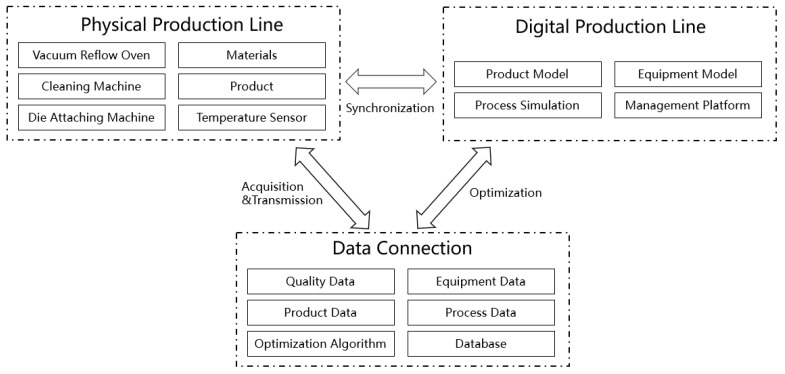
Real time optimization process for intelligent production line.

**Figure 14 sensors-24-00612-f014:**
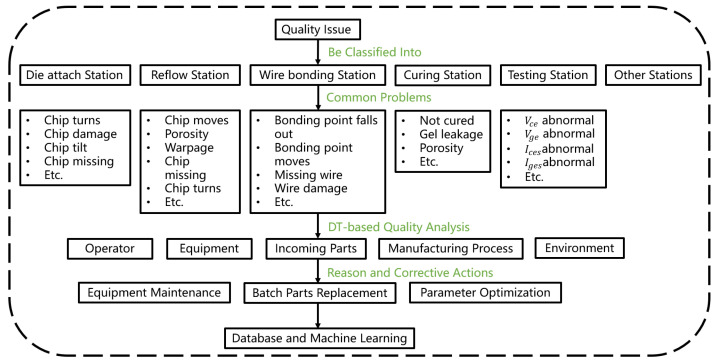
Digital-twin-driven quality issue solving method.

**Figure 15 sensors-24-00612-f015:**
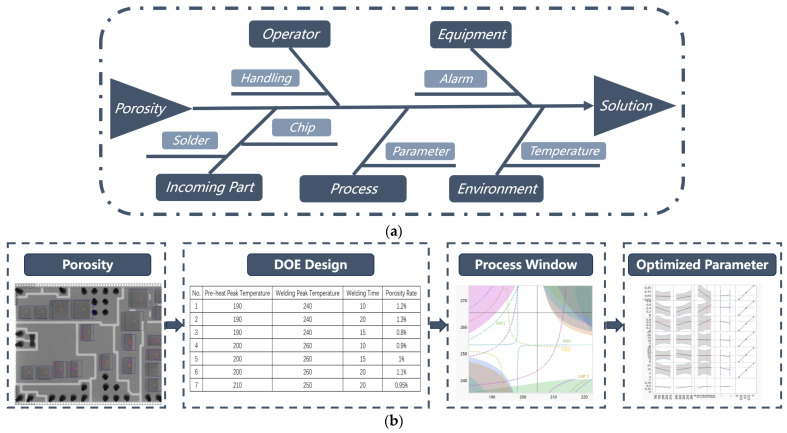
Porosity quality issue solving: (**a**) Digital-twin-based quality analysis frame. (**b**) Parameter optimization for the reflow process.

**Figure 16 sensors-24-00612-f016:**
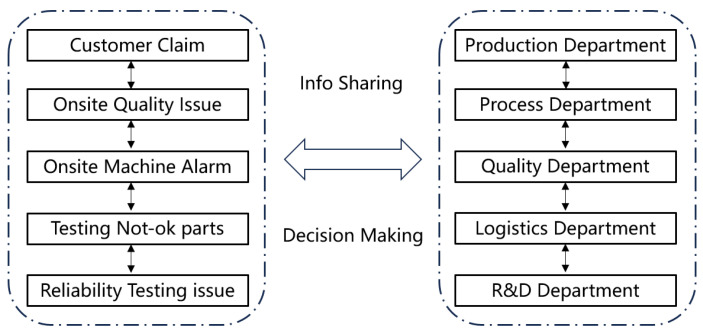
Real-time information sharing among various departments.

**Figure 17 sensors-24-00612-f017:**
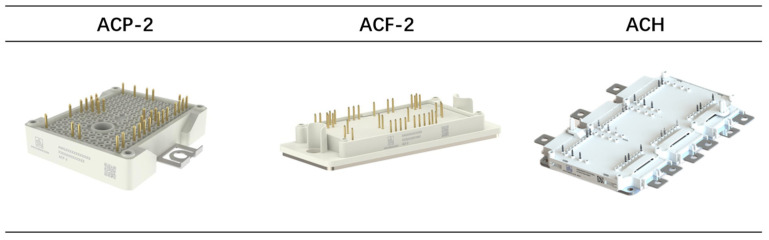
Three products manufactured by an intelligent production line.

**Table 1 sensors-24-00612-t001:** Main equipment and sensors for IGBT production line.

Equipment	Acquisition	Data	Data Type
Screen printing	Displacement sensor	Solder paster thickness	Float
Conveyor belt	Vision sensor	Position info	Float
Die attaching	Vision sensor	Die position	Float
Vacuum reflow	Temperature sensor	Temperature	Float
Wire bonding	Force sensor	Force	Float
Injection molding	Weight sensor	Weight	Float

**Table 2 sensors-24-00612-t002:** Information of three types of typical IGBT modules.

Type	$V_{CES}$ (V)	$I_{C}$ (A)	Application	$E_{on}$ (mJ)	$E_{off}$ (mJ)	$R_{th}$ (K/W) (Before)	$R_{th}$ (K/W) (After)	Product Yield (Before)	Product Yield (After)
ACP-2	650	100	3-level	0.84	1.51	0.85	0.71	93%	96%
ACF-2	650	450	3-level	8.18	10.36	0.45	0.35	90%	94%
ACH	650	820	3-level	15.7	22.5	0.24	0.11	92%	95%

**Table 3 sensors-24-00612-t003:** Comparison between the traditional production line and the digital-twin-based production line.

Format of Production Line	Working Mode	Information Acquisition	Optimization	Product Yield
Traditional IGBT production line	- Paper process	- Personal communication	- Experiment iteration	≈90%
	- Paper record	- Meeting	- By experience	
	- Paper work instruction	- Fragmented information	- Time-spending	
Digital-twin-based IGBT production line	- Digital database	- Information management system	- Optimize algorithm	≈95%
	- Digital working instruction	- Real-time information	- Simulation	
	- Digital info sharing	- Complete info-sharing	- Time-saving	

**Table 4 sensors-24-00612-t004:** Application of digital-twin technology in other industries.

Application Field	Key Enabling Technologies	Advantages	Product	Performance
Apparel Manufacturing	- Data collection, incorporation	- Efficient production	\	- Improved output performance
	- Realtime decision making	- Meet individualized production requirement		
	- Line balancing			
FPCB Etching production Line [41]	- Data acquisition	- Realtime adjustment of process parameters	YES	- Improved product yield
	- Realtime adjustment of process parameters			
	- Process simulation data	- High precision and efficiency		
Paper Manufacturing	- Prediction model for key process:	- Parameter optimization	\	- Improved production efficiency
	- Stirring speed model of the dump chest	- Artificial intelligence		- Cost saving
	- Deflaker water consumption model			
	- Supply air pressure model			
Manual Assembly Line	- Progressive multidimensional DT modeling method	- Real-time synchronization and monitoring	\	- Shorter unit order completion time
	- Simulation system function	- High accuracy		
	- Virtual system function			
Catalyst Manufacturing Line [42]	- Machine learning-based DT	- Monitor process parameters	YES	- \
	- Data collection			
	- Data preprocessing	- Accuracy		
	- Predictive modeling			
IGBT production line (this work)	- Process simulation database	- Modular production	YES	- Improved product yield
	- Realtime monitoring	- DT-based quality issue solving		- Better product characteristic performance
	- Information management system	- Process parameter optimization		

## Data Availability

Data available on request from the authors.

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
