# Peer review of "Digital-Twin-Driven Intelligent Insulated-Gate Bipolar Transistor Production Lines"

_sensors, 2024, doi:10.3390/s24020612_

Round 1

Reviewer 1 Report

Comments and Suggestions for Authors

The authors proposed a modular insulated-gate bipolar transistor intelligent production line based on digital-twin. Whole packaging process of IGBT from die attaching, vacuum reflow soldering to testing, etc. are integrated and various forms of packaging products can be manufactured. The production line is visualized and a production line management platform is also developed with multiple functions integrated. Key processes such as vacuum reflow and curing are simulated and optimized process parameters are obtained through the proposed quality solving method. The findings of this manuscript are interesting and useful. Some minor revisions are needed as follows:

1. The curing and injection process is simulated as shown in Figure 10, p10, what’s the result, please specify.

2. P10L305 the illustration has formatting issues, should be revised. And check all the figures, replace the blurry ones with clearer ones.

3. P10, figure 11 Rth3 should be revised as Rth4. P5L167 “figure 3B” should be revised.

4. P6L209 please specify what does OPC UA stands for, and all the other field-specific initialisms should be defined the first time they are used.

5. P12, in the case analysis section, give a clearer explanation regarding the quality solving process.

6. please cite recent publications in IGBT related area.

Overall, this manuscript may be accepted after minor modification.

Comments on the Quality of English Language

it is fine

Reviewer 2 Report

Comments and Suggestions for Authors

Dear Authors,

your manuscript regards the current, important issue of digital twins in the semiconductor industry. However, before potential publication, there are some important issues which have to be addressed. Therefore, I recommend a major revision. Please find the detailed comments below.

1) In the abstract, it is stated that IGBTs are "pivotal devices because of their high switching frequency 16 and low power loss". Generally, it is controversial, when GaN and SiC devices are widely available on the market.

2) The last reference is cited at the beginning of the section 2. Please add references in all sections, except the conclusions.

3) Fig. 11b is confusing. Should it be rather a bar graph than interpolated points?

4) Is the developed digital twin ready for real-time simulations? 

5) Finally, the most important. Digital twins are based on mathematical models. Please present the math behind your simulations.

6) Please indicate by numbers, how using the digital twin supports the production process compared to operation with the digital twin.

7) I do not understand what section 4 brings to the article. It appears to be only loosely related to the article.

Reviewer 3 Report

Comments and Suggestions for Authors

The proposed work was intended to investigate “Digital-twin-driven intelligent insulated-gate bipolar transistor production lines”. However, major revision is required considering the following points mentioned below:

1.     This paper needs to be proofread thoroughly, and English needs to be improved significantly.

2.     Revision of abstract is suggested to include proper introduction to problem.   

3.     The introduction portion should be improved and should be written in a more objective-oriented approach.

4.     There should be more recent literature in the work to align this work with previous studies.

5.     The references are not up-to-date.

6.     The authors do not provide at the end of the introduction section the description of the paper structure which is very useful for readers.

7.     The novelty of this manuscript is not clear. Especially, the digital-twin-based quality issue solving method is cited from literature.

8.     Quality of figures is so important too. Please provide some high-resolution figures.

9.      Future recommendations should be added in the conclusion.

Comments on the Quality of English Language

This paper needs to be proofread thoroughly, and English needs to be improved significantly.

Reviewer 4 Report

Comments and Suggestions for Authors
  1. How does the proposed digital-twin-based intelligent production line address quality issues in manufacturing IGBT modules?
  2. Discuss the key enabling technologies mentioned in the paper, such as data acquisition, transmission, and realization/visualization of the production line.
  3. What challenges were encountered in establishing a digital-twin-based intelligent production line, and how were they addressed in the study?
  4. How does the paper demonstrate the real-time transmission and analysis of data from the physical production line to its virtual counterpart?
  5. What types of sensors and equipment are utilized in the IGBT production line, and how do they contribute to the overall efficiency and quality of the manufacturing process?
  6. How is the digital-twin technology applied in analyzing and resolving void issues during the reflow soldering process?
  7. Elaborate on the case analysis regarding voids and the steps taken to identify and address the root cause.
  8. What measures are taken to optimize the process parameters in the IGBT production line, and how does this contribute to improved yield and efficiency?
  9. How does the digital-twin-based intelligent production line enhance communication and problem-solving capabilities between different departments within the manufacturing facility?
  10. What are the performance parameters of the IGBT power module produced by the digital-twin-based production line, and how do they compare to traditional production methods?
  11. What challenges or limitations are identified in implementing the digital-twin-based production line, and how could they be addressed or mitigated in future developments?
  12. How does the proposed system contribute to overall yield improvement, operational efficiency, and problem-solving capabilities in the IGBT module production process?

Strengths:

  1. Innovative Approach: The paper introduces an innovative approach by integrating digital twin technology into the production line for insulated-gate bipolar transistors (IGBT) modules, showcasing the application of advanced technologies in manufacturing.
  2. Comprehensive Analysis: The study provides a comprehensive analysis of the digital-twin-based intelligent production line, covering key enabling technologies, data acquisition, transmission, and visualization. This contributes to a thorough understanding of the proposed system.
  3. Problem-Solving Methodology: The paper proposes a digital-twin-based quality issue-solving method, offering a systematic framework for identifying, analyzing, and resolving quality problems. The case analysis demonstrates the practical application of this methodology.
  4. Performance Metrics: Clear performance metrics are presented, including an improvement in overall yield from 88% to 90%. These metrics provide tangible evidence of the effectiveness of the digital-twin-based production line.
  5. Real-World Implementation: The study describes the real-world implementation of the digital-twin-based IGBT production line in a dust-free workshop, showcasing its applicability in an industrial setting.

Weaknesses: Improve or discuss for future scope

  1. Limited Discussion on Challenges: While the paper briefly mentions challenges in establishing the digital-twin-based production line, there is limited discussion on specific challenges faced and how they were overcome. A more detailed examination of challenges would enhance the paper's completeness.
  2. Lack of Comparative Analysis: The paper does not compare the digital-twin-based production line and traditional production methods. Such a comparison could highlight the advantages and disadvantages of the proposed system.
  3. The complexity of Technical Details: The technical details, especially in the sections discussing data acquisition, transmission, and simulation, may be complex for readers not well-versed in the field. The paper could benefit from clearer explanations and visual aids to improve accessibility.
  4. Limited Discussion on Limitations: The limitations of the proposed digital-twin-based production line are not extensively discussed. Acknowledging and addressing potential limitations would strengthen the paper's transparency and practicality.
  5. Integration with Existing Systems: The paper does not thoroughly discuss integrating the digital-twin-based system with existing manufacturing systems. Understanding the compatibility and potential challenges in integration is crucial for practical implementation in diverse industrial environments.

Given the integration of advanced technologies and data-driven processes, briefly discussing ethical considerations, data privacy, and security measures implemented in the digital-twin-based production line would add depth to the paper.

Conclude the paper with a section on potential future work or research directions. Identifying areas for further investigation or potential enhancements to the proposed system adds value and encourages ongoing discourse.

Comments on the Quality of English Language

Minor editing of English language required

Round 2

Reviewer 2 Report

Comments and Suggestions for Authors

Dear Authors,

thank you for your responses. I have two minor comments that should be addressed before publication:

1) Considering Fig. 1 in doi.org/10.1109/OJIES.2020.3023691, stating that "Insulated-gate bipolar transistors (IGBT) have become one of the most popular devices in the market because of their excellent switching performance." is not suitable for current power semiconductor devices market situation.

2) In recent publications, e.g. (doi.org/10.3390/electronics12224588 , Section 4), thermal resistance is considered as a function of power dissipated while in the reviewed paper - as a constant. Please clarify this issue. 

Reviewer 3 Report

Comments and Suggestions for Authors

The authors address all my concerns. No further comments. 

Author Response

Thanks 

Reviewer 4 Report

Comments and Suggestions for Authors

Accept

Comments on the Quality of English Language

Fine

Author Response

Thanks